

# Building extraction from remote sensing images based on multi-scale attention gate and enhanced positional information

Rui Xu[1],[*], Renzhong Mao[1],[*], Zhenxing Zhuang[1], Fenghua Huang[2] and Yihui Yang[1]

[1] School of Computer Science and Mathematics, Fujian University of Technology, Fuzhou, Fujian, China
[2] Fujian Key Laboratory of Spatial Information Perception and Intelligent Processing (Yango University), Fuzhou, Fujian, China
* These authors contributed equally to this work.

Corresponding authors
Renzhong Mao,
mao_renzhong@126.com
Fenghua Huang,
fhhuang@ygu.edu.cn

## ABSTRACT

Extracting buildings from high-resolution remote sensing images is currently a research hotspot in the field of remote sensing applications. Deep learning methods have significantly improved the accuracy of building extraction, but there are still deficiencies such as blurred edges, incomplete structures and loss of details in the extraction results. To obtain accurate contours and clear boundaries of buildings, this article proposes a novel building extraction method utilizing multi-scale attention gate and enhanced positional information. By employing U-Net as the main framework, this article introduces a multi-scale attention gate module in the encoder, which effectively improves the ability to capture multi-scale information, and designs a module in the decoder to enhance the positional information of the features, allowing for more precise localization and extraction of the shape and edge information of buildings. To validate the effectiveness of the proposed method, comprehensive evaluations were conducted on three benchmark datasets, Massachusetts, WHU, and Inria. The comparative analysis with six state-of-the-art models (SegNet, DeepLabv3+, U-Net, DSATNet, SDSC-Unet, and BuildFormer) demonstrates consistent performance improvements in intersection over union (IoU) metrics. Specifically, the proposed method achieves IoU increments of 2.19%, 3.31%, 3.10%, 2.00%, 3.35%, and 3.48% respectively on Massachusetts dataset, 1.26%, 4.18%, 1.18%, 2.01%, 2.03%, and 2.29% on WHU dataset, and 0.87%, 5.25%, 2.02%, 5.55%, 4.39%, and 1.18% on Inria dataset. The experimental results indicate that the proposed method can effectively integrate multi-scale features and optimize the extracted building edges, achieving superior performance compared to existing methodologies in building extraction tasks.

# INTRODUCTION

With the rapid development of high-resolution sensor technology, the spectral and spatial resolution of images has been greatly improved, which makes high-resolution remote

sensing images easily accessible and provides new possibilities for the application of remote sensing technology. Extracting real-time and accurate building information from remote sensing images provides essential data for urban planning (*Fan et al., 2022*), ecological monitoring (*He et al., 2022*), map production (*Vo et al., 2013*) and so on. An increasing number of high-resolution remote sensing images provide richer information about the Earth's features (*Shao et al., 2020*), but also pose unprecedented challenges for the effective use of remote sensing data (*Huang et al., 2024*). Many factors can lead to missed and false detections in the extraction results, such as the diversity in building structures and shapes, spectral characteristics, numerous surrounding disturbances and complex image backgrounds. How to accurately and effectively extract buildings from remote sensing images remains an urgent challenge that needs to be addressed.

Traditional building extraction methods mainly include edge feature method, region segmentation method, semantic classification method, and prior knowledge model method, *etc*. The edge feature method (*Lu et al., 2018*) identifies building contours by detecting edge continuity. While effective in simple environments, its heavy reliance on edge information makes it susceptible to errors in complex image backgrounds. The region segmentation method (*Müller & Zaum, 2005*) identifies buildings by segmenting areas based on pixel similarity, which is effective for images with distinct color and texture differences, but performs poorly when the spectral characteristics of the buildings do not significantly differ from the background. The semantic classification method (*Li et al., 2015*) uses machine learning for semantic classification of pixels or regions, which can handle complex scenes but requires a large amount of annotated data. The prior knowledge model method (*Shi et al., 2016*) constructs a model based on the characteristics of known buildings, which is effective in specific image scenarios but has limited generalization capability. The afore-mentioned building extraction methods mainly map the low-level features of images to semantic features, relying on researchers' understanding and experience of low-level features. While demonstrating satisfactory performance in controlled environments, these methods frequently exhibit critical limitations when processing images containing intricate background interference, manifesting as fragmented building outlines, false positives, omissions, and incomplete feature extraction in heterogeneous landscapes.

In recent years, deep learning methods have attracted increasing attention and have been widely applied in areas such as object detection (*Chen et al., 2024*), image classification (*Zheng, 2024*), and semantic segmentation (*Wu et al., 2024*). *Long, Shelhamer & Darrell (2015)* applied convolutional neural networks (CNNs) to building extraction, which significantly improved the extraction accuracy compared to traditional methods. The fully convolutional networks (FCNs) designed by *Long, Shelhamer & Darrell (2015)* achieved end-to-end pixel-level segmentation by replacing fully connected layers with convolutional layers. Since then, a series of network models based on the encoder-decoder structure have been proposed, such as Segnet (*Badrinarayanan, Kendall & Cipolla, 2017*), Pspnet (*Zhao et al., 2017*), ResNet (*He et al., 2016*), Deeplab (*Chen et al., 2018*), U-Net (*Peng et al., 2021*), all of which have revealed both technical innovations and inherent limitations. *Abdollahi, Pradhan & Alamri (2022)* used Segnet for building extraction,

where the decoder performs non-linear upsampling using indices calculated at the max-pooling layer of the encoder, enhancing the effect of building boundary segmentation. However, the decoder's architectural constraints exhibit significant limitations in reconstructing intricate structural details when processing geometrically complex buildings or scenes with dynamic environmental variations, particularly under multi-scale spatial configurations. *Chen et al. (2021)* combined dense connections with residual networks in Deeplabv3+ to alleviate the incomplete integration of high and low-level features of buildings and improve the accuracy of building extraction, but excessive feature fusion may lead to overfitting of the model, thereby affecting its generalization ability and robustness. To better model long-distance dependencies (*Liu et al., 2021*), the self-attention mechanism has been introduced into image segmentation (*Pu et al., 2024*). *Wang et al. (2022)* designed a building extraction network Buildformer, which employs a multi-head self-attention mechanism and a global context path module to capture global dependencies. While excelling in global dependency modeling, the framework's prioritization of global features compromises local feature resolution, particularly affecting precision on small-scale structures and boundary-indistinct buildings. *Wan et al. (2023)* designed a feature refinement module in the channel of DSAT-Net to fuse low-level and high-level features. These asymmetric structure networks usually adopt a simplified design in the decoder to reduce the computational cost associated with the complex results from the encoder, but the simplified design may lead to the decoder's inability to effectively restore detailed information when restoring spatial resolution.

The U-Net network is a symmetric encoder-decoder structure (*Peng et al., 2021*), which directly transmits the features of each layer of the encoder to the corresponding layer of the decoder through skip connections, enabling the decoder to effectively utilize the low-level detail information from the encoder. Due to its advantages, such as its simplicity, flexibility, fewer parameters, and strong detail-capturing ability, the U-Net network has been applied into building extraction and improved its performance. For example, *Yu et al. (2023)* introduced recursive, residual deformable convolution units into the U-Net structure, enhancing the model's ability to learn complex details of buildings. *Xu et al. (2023)* used a feature pyramid in the decoder of U-Net to fuse feature maps of different scales in order to extract more building features from images. *Jin et al. (2021)* integrated dense spatial pyramid pooling (*Yang et al., 2018*) into the U-Net structure to refine the boundaries perception of large building, enhancing the integrity of large building segmentation. *Zhang, Zhang & Zhang (2023)* introduced a novel branching transformer in SDSC-UNet, enabling the model to capture multi-scale information internally while fully establishing global dependency relationships.

The afore-mentioned U-shaped methods with symmetric encoder-decoder structures have achieved good extraction results. However, they still have shortcomings in accurately capturing image features and contextual information of buildings at different scales. Especially when extracting buildings from remote sensing images with complex backgrounds, these methods often encounter issues such as false positives, missed detections, and incomplete edges. Their main deficiencies can be summarized in two aspects. First, the convolution and pooling operations during the upsampling process can

lead to the loss of spatial feature information. Second, these network structures lack the ability to capture building location information during the fusion of high-level and low-level features, resulting in poor extraction of building detail information by the model. To overcome these shortcomings, this article designs a novel method for building extraction based on the U-Net framework, which improves the network at both the encoding and decoding stages.

In the encoding stage, a multi-scale attention gate module is introduced which effectively integrates contextual information by fusing building features of different scales, while suppressing background noise interference.

In the decoding stage, the traditional U-Net upsampling method is improved by adding a dual-path upsampling which retains more high-level semantic information in feature maps of different scales. During the upsampling process, positional information is embedded into channel attention to enhance the positional information of features. The improved network not only captures cross-channel information but also incorporates directional and position-aware information, allowing for more accurate localization and extraction of building shapes and edge details. This optimization helps refine building boundaries and corner details in complex background images, thereby enhancing the model's performance.

# METHOD

## Network structure

UNet extracts and decodes features from a single dimension, which limits its ability to capture the spatial and positional characteristics of multi-scale information and prevents it from effectively utilizing multi-scale features across different levels of abstraction. Therefore, this article designs a network model enhanced with multi-scale attention gate and positional information.

The network structure is shown in Fig. 1. The network accepts an input image with a height of H, a width of W, and three channels. The stage is the invariant phrase in U-Net. After passing through the first five stages, the image's width and height are reduced by half, while the number of channels increases. The multi-scale attention gate mechanism captures multi-scale information during the downsampling process, extracting multi-scale information to be fed into the next downsampling and skip layer connections. In the decoder part, improved upsampling is used to obtain information between different feature layers, standardizing the size of feature maps at different scales. Then, through skip layer connections, the upsampled feature maps are fused with the feature maps processed by the multi-scale attention gate mechanism, enhancing the positional information, and progressively decoding the predicted feature maps to achieve the final segmentation results.

## Multi-scale attention gate module

The overall structure of the multi-scale attention gate is shown in Fig. 2. Three types of convolutions with different receptive fields are introduced to extract features: pointwise convolution (*Sandler et al., 2018*), regular convolution (*i.e.*, kernel size of $3 \times 3$, stride of 1,

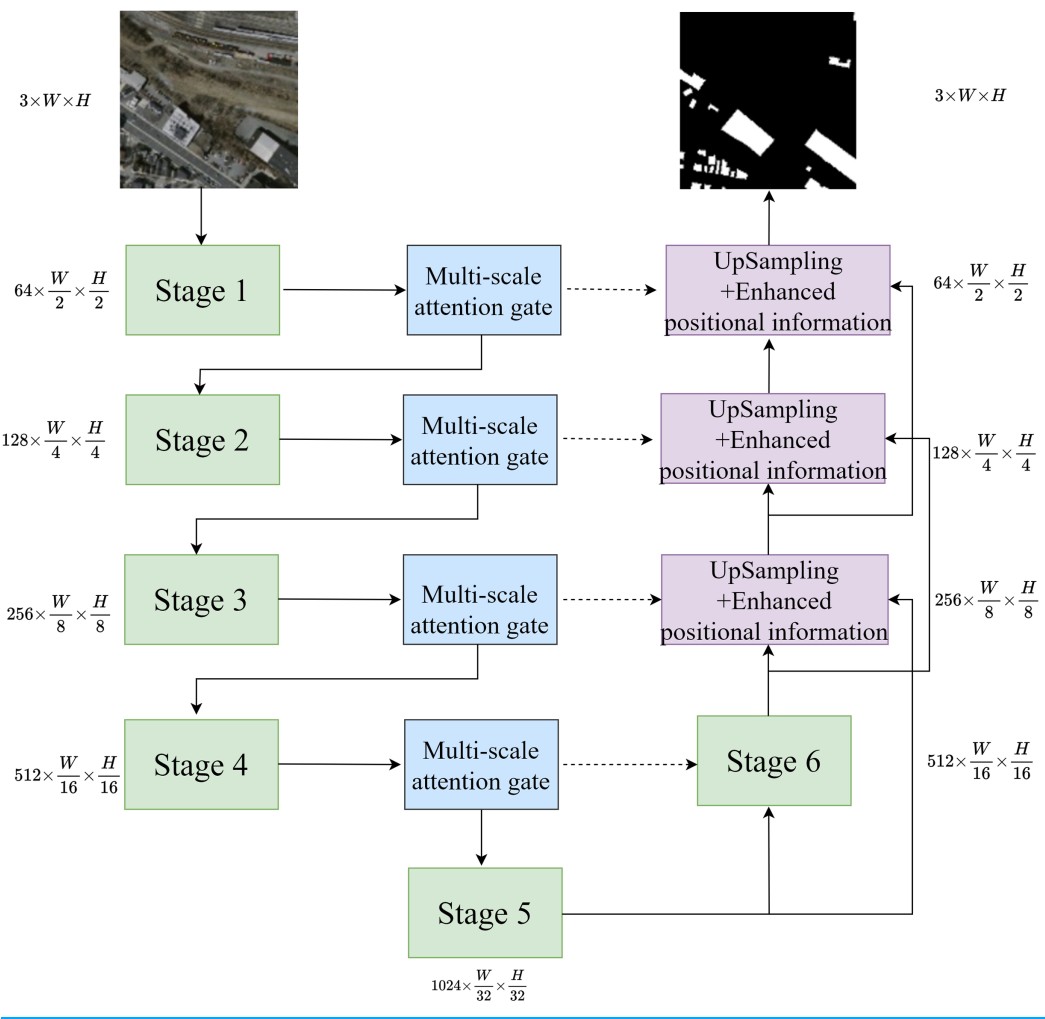

**Figure 1 Network structure.**

and padding of 1), and dilated convolution (*i.e.*, kernel size of 3 × 3, stride of 1, padding of 2, and dilation rate of 2) (*Li et al., 2022*). The convolutions in the three parallel branches provide different receptive fields, effectively extracting features from buildings at different scales, thereby enabling the model to capture multi-scale features (*Nguyen & Nguyen, 2023*). Among them, pointwise convolution uses 1 × 1 convolutional kernels to fuse features across channels, which can finely extract detailed textures in buildings and effectively focus on small-scale buildings. Ordinary convolution obtains information about buildings at regular scales. A padding of 1 means adding additional pixels to the edges of the input feature map which allows the convolution kernel to fully cover the input image's edge regions. After convolution, the spatial size of the output feature map remains unchanged, preserving the edge information. A stride of 1 means that the convolution kernel moves 1 pixel each time for convolution operation, enabling detailed feature extraction. The dilated convolution has a dilation rate of 2. By adding holes to expand the receptive field of the convolution, the convolution operation can cover a larger area. Due to its larger receptive field, the network can capture building information at a larger scale.
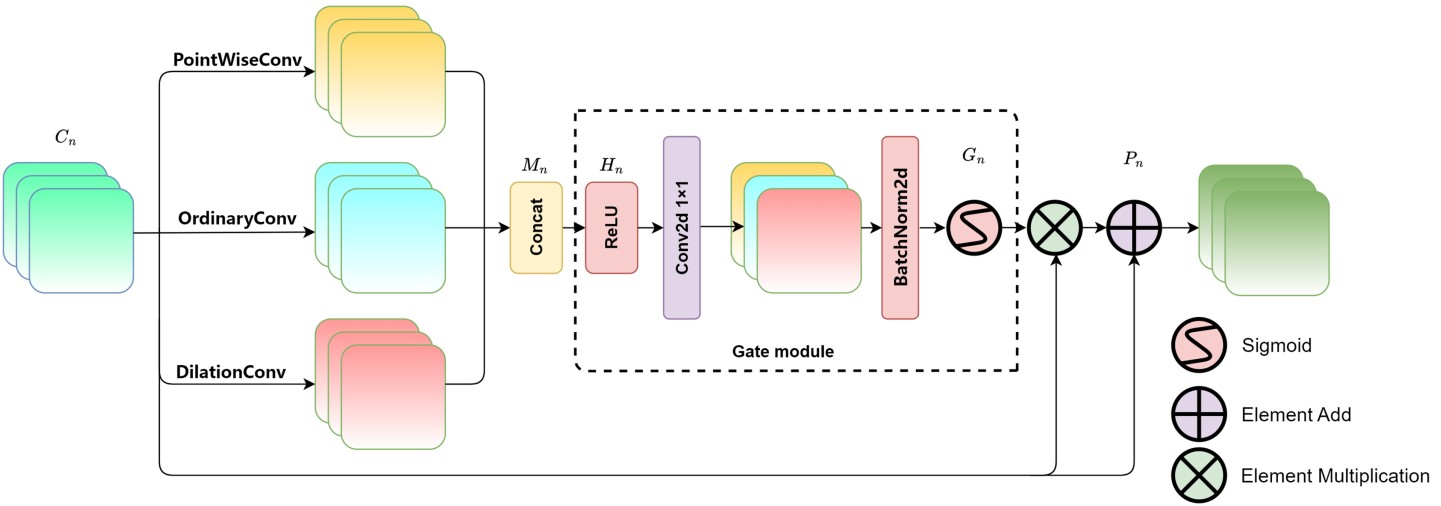

**Figure 2 Multi-scale attention gate module.**

The gating mechanism (*Dong et al., 2022*) is a mature adjustment strategy, which was originally designed to control the flow of information to measure the importance of each feature vector. This article concatenates and fuses the feature maps of the original image after being processed by three different scales of convolution, and then the fused result is input in a gating unit for adaptive adjustment of multi-scale information, enhancing the model's final feature representation capability. The feature maps processed by multi-scale convolution are denoted as, $M_n$ which serve as the input for feature fusion. The fusion formula is as follows:

$$M_n = concat\left(Conv_p(C_n) + Conv_o(C_n) + Conv_d(C_n)\right) \tag{1}$$
$$H_n = Conv_{1\times1}(relu(M_n)) \tag{2}$$
$$G_n = \text{Sigmoid}(\text{Bn2d}(H_n)) \tag{3}$$
$$P_n = C_n \times (G_n + 1). \tag{4}$$

Considering the specific definition of convolution kernel size in the formula, the relevant symbol interpretation table is listed in Table 1. In the formula, $n$ represents the current feature level ($n \in \{1, …,l\}$) and $l$ represents the model hierarchy. $C_n$ is the feature map from the encoding layer. $Conv_p$ refers to the pointwise convolution, $Conv_o$ is the regular convolution, and $Conv_d$ is the dilated convolution. $H_n$ is the mixed feature map. $G_n \in (0, 1)$ represents the gating output. $Conv_{1\times1}$ is the regular convolution with a kernel size of $1 \times 1$, and $P_n$ is the fusion result. According to Eqs. (1) and (2), the encoding layer feature map $C_n$ undergoes convolutions through three parallel branches and is then concatenated into a multi-scale feature map $M_n$, which is then input into the gating unit. $M_n$ is processed through ReLU activation and a $1 \times 1$ convolution, resulting in the initial fusion of multi-scale features and generating the mixed feature map $H_n$. As indicated by Eq. (3), $G_n$ is obtained by normalizing $H_n$ using batch normalization and applying the sigmoid activation function, mapping the output weights between (0, 1). According to

**Table 1 Explanation table of related symbols.**

| Symbol | Explain |
|---|---|
| $n$ | Current feature level, $n \in \{1,\ldots,l\}$, representing the level number |
| $C_n$ | Encoding layer feature map, representing the feature map of the nth layer |
| $Conv_p$ | Pointwise convolution, kernel_size = 1, padding = 0 |
| $Conv_o$ | Ordinary convolution, kernel_size = 3, padding = 1, stride = 1 |
| $Conv_d$ | Dilation convolution, kernel_size = 3, padding = 2, stride = 1, dilation = 2 |
| $Conv_{1 \times 1}$ | $1 \times 1$ convolution, kernel_size = (1, 1), used for dimensionality reduction |
| $H_n$ | Mixed feature map after dimensionality reduction |
| Bn2d | Batch Normalization 2D, methods for standardizing input |
| $G_n$ | Gate controlled output, $Gn \in (0, 1)$ |
| $P_n$ | The final processed feature image |

Eq. (4), $G_n$ is multiplied element-wise with $C_n$ enhancing the multi-scale features. Finally, it is added to the original image $C_n$ to retain the detailed features from the original image, resulting in the output feature image $P_n$ with multi-scale information guidance. The multi-scale attention gate module captures the multi-scale features of buildings by adaptively learning the weights at each spatial position, thereby improving the accuracy of the extraction.

## Positional information enhancement module

In the process of decoding the multi-scale feature maps, single-layer upsampling can lead to the loss of spatial feature information, making it difficult to retain a large amount of high-level semantic information. Skip connections simply concatenate all channel features without focusing on different channels, resulting in insufficient contextual information. In this module, a dual-path upsampling approach is used to obtain detailed features and high-level semantic information from different feature layers. Then, the acquired features undergo positional attention processing to enhance their positional information.

The overall structure is shown in Fig. 3. The input end accepts the $f_{MSAG}$ processed by the multi-scale attention gate module from the encoding layer and the $f_{up}$ from the dual-path upsampling, which are input to the positional information module for processing after concatenation and convolution. Positional information is embedded along the horizontal X-axis and the vertical Y-axis. Finally, the processed feature map is superimposed on the original image to obtain the final output $f_{out}$.

### Dual path upsampling

The structure of the dual-path upsampling module is shown in Fig. 4. This module performs a two-fold upsampling on the feature map $f_i$ of the decoding layer at level $i$ and a four-fold upsampling on the feature map $f_{i-1}$ at level $i$-1, using bilinear interpolation. Convolutional normalization is performed separately, with a convolution kernel size of $3 \times 3$, to restore the feature maps to the same size. Finally, the feature maps are added together to produce the output feature map.

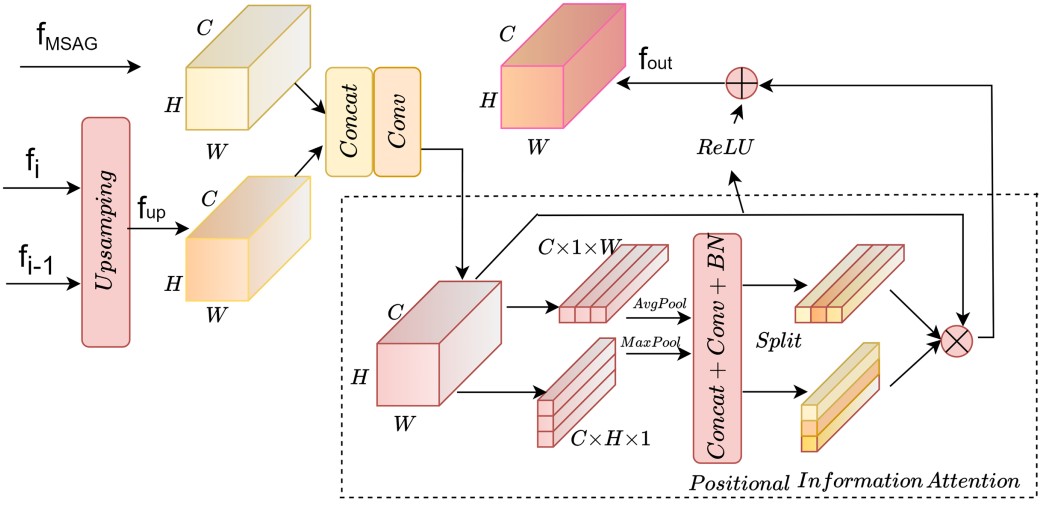

**Figure 3 Position information enhancement module.**

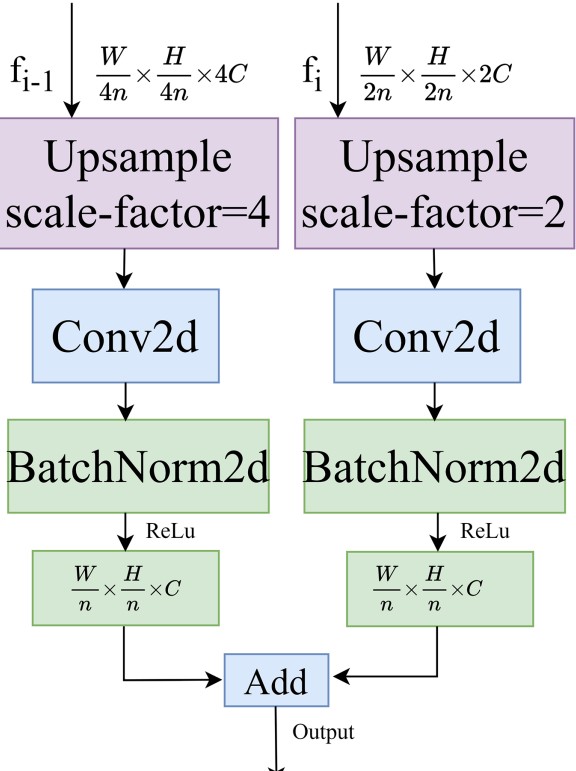

**Figure 4 Dual path upsampling.**

## Positional information attention

To enable the model to obtain sufficient spatial location information, positional information attention is used to further process the features obtained by upsampling. Regular convolution has difficulty modeling channel relationships, while pooling helps the model capture global information, which perfectly compensates for the shortcomings of

convolution. As shown in Fig. 5A, the SE attention mechanism (*Hu, Shen & Sun, 2018*) directly applies global average pooling to the feature maps to obtain a C × 1 × 1 dimensional vector, compressing the spatial information into one dimension. It only considers re-evaluating the importance of each channel by modeling channel relationships, but ignores positional information. As shown in Fig. 5B, the CBAM attention mechanism (*Woo et al., 2018*) sequentially processes through channel attention and spatial attention, multiplying the output of the channel attention (1D convolution) element-wise with the spatial attention (2D convolution). This allows it to learn local relationships, but it is difficult to learn long-range dependencies.

Therefore, in this article, positional information is embedded into the channel attention mechanism, enabling the network to focus on positional information over a larger area. To mitigate the loss of positional information caused by two-dimensional global pooling, the channel attention is decomposed into multiple parallel one-dimensional feature encoding processes. Global average pooling and max pooling are performed using two kernels of sizes (H, 1) and (1, W). The processing steps are shown in Eqs. (5)–(8), where $x$ represents the input features, represents the output after average pooling, and $m$ represents the output after max pooling.

Equations (5) and (6) represent the average pooling and max pooling, respectively, for the c-th channel at height h:

$$a_c^h(h) = \frac{1}{w} \sum_{i < w} x_c(n, i) \tag{5}$$

$$m_c^h(h) = max \sum_{i < w} x_c(n, i). \tag{6}$$

Equations (7) and (8) represent the average pooling and max pooling, respectively, for the c-th channel at width w:

$$a_c^w(w) = \frac{1}{h} \sum_{i < h} x_c(n, i) \tag{7}$$

$$m_c^w(w) = max \sum_{i < h} x_c(n, i). \tag{8}$$

Then, the feature maps embedded with specific directional information are fused through concatenation, 1 × 1 convolution, normalization, and activation processing sequentially. $F_1$ represents the 1 × 1 convolution process, as shown in Eq. (9):

$$f = ReLu\left(Bn2d(F_1 \left[a^h, m^h, a^w, m^w\right])\right). \tag{9}$$

Finally, the output features $f$ are split along the X and Y axes for convolution and activation. Each element in the two resulting attention maps reflects whether a building exists in the corresponding row and column. Two 1 × 1 convolutions, $F_h$ and $F_w$, are used to transform the output $f$ into tensors of C × H × 1 and C × 1 × W, respectively, with outputs $F_h$ and $F_w$ shown in Eq. (10):

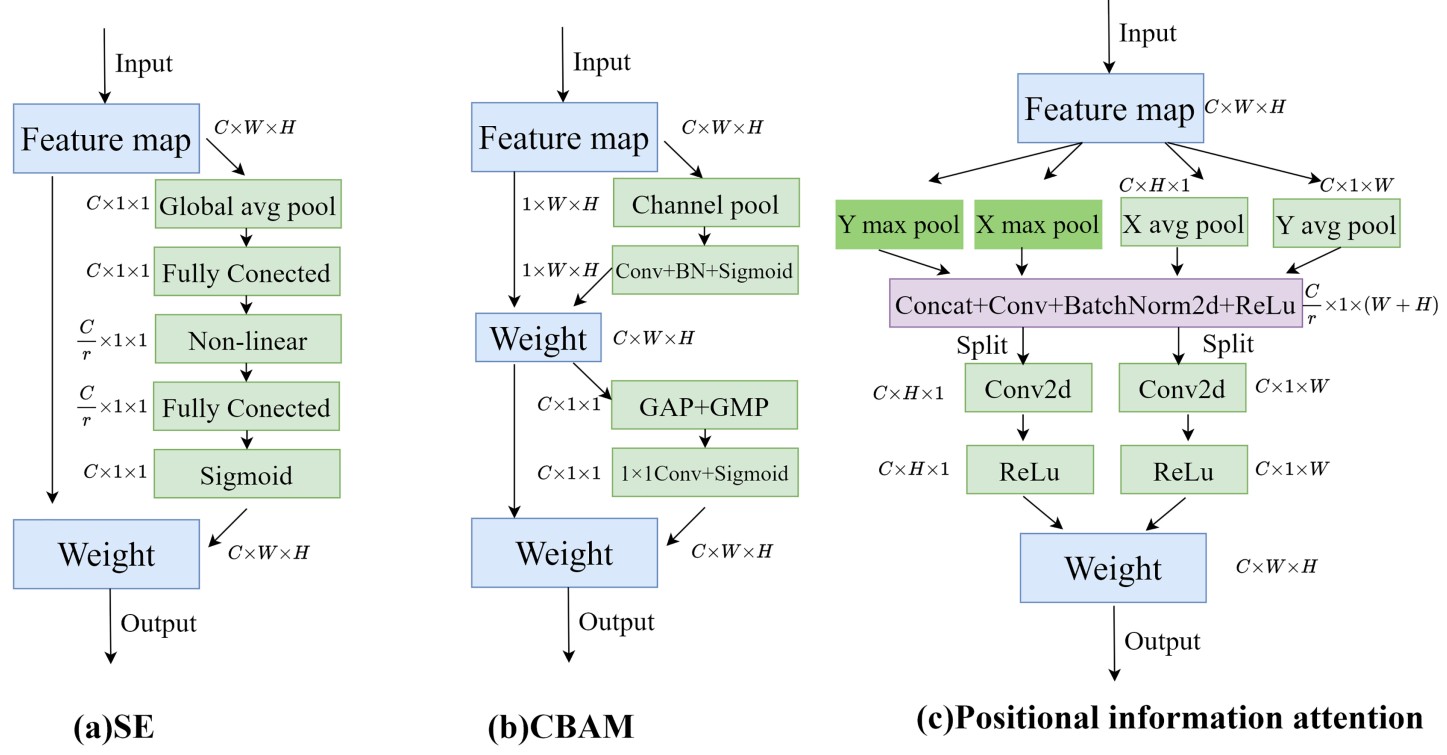

**Figure 5 Comparison of attention modules; (A) SE attention; (B) CBAM attention; (C) Position information attention.**

$$f_h = ReLu(F_h(f)), \ f_w = ReLu(F_w(f)). \tag{10}$$

# EXPERIMENT

## Training environment

The experiments in this article are based on the Linux Ubuntu system environment, with one 2080Ti 11GB graphics card used on the platform. On this basis, the Pytorch deep learning framework is adopted for the design and construction of the model, using the GPU computing platform with CUDA version 11.7.

The Adam method from the adaptive learning rate gradient descent algorithm is used to optimize the loss function, with an initial learning rate of $1e^{-4}$ and minimun learning rate of $1e^{-6}$. The cosine strategy were employed. To ensure training stability and model convergence, the training batch size of the network is set to 2, and the number of training iterations is set to 100 rounds. As shown in Fig. 6, the loss decreases rapidly during the first 20 rounds of training. As the number of iterations increases, the loss gradually stabilizes, reaching near-convergence within 100 rounds.

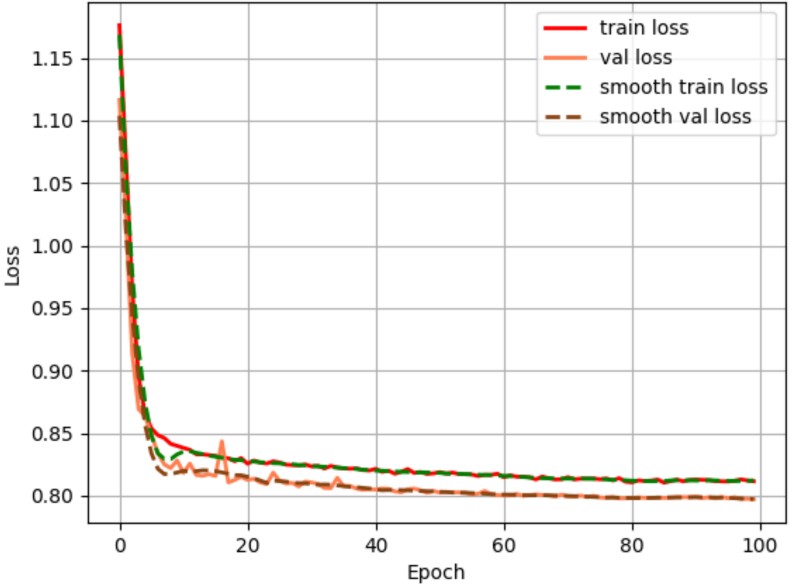

**Figure 6 Loss convergence plot.**

## DATASET

This article employs three public datasets for experimental validation, the Massachusetts Building Dataset (https://www.cs.toronto.edu/~vmnih/data/, accessed on 10 December 2023), the WHU Building Dataset from Wuhan University (http://gpcv.whu.edu.cn/data/building_dataset.html, accessed on 15 January 2024), and the Inria Aerial Image Labeling Datase (https://project.inria.fr/aerialimagelabeling/, accessed on 12 January 2024).

The Massachusetts Building Dataset covers various types of buildings in urban and suburban areas of Boston, USA, such as office buildings, residential homes, and garages. The dataset includes 151 high-resolution remote sensing images sized at 1,500 pixels × 1,500 pixels with a resolution of 1.0 m, covering an area of approximately 340 km$^2$ on the ground. The images were uniformly cropped to a size of 256 × 256, resulting in 5,796 images. Out of these, 4,060 images were selected for the training set, 360 images for the validation set, and 1,376 images for the test set.

The WHU Building Dataset from Wuhan University is a large dataset composed of multi-source remote sensing images, primarily including aerial and satellite remote sensing imagery, with each image sized at 512 × 512. There are a total of 8,189 aerial images with a spatial resolution of approximately 0.3 meters, covering a ground area of about 450 km$^2$. The images are pixel-wise annotated as buildings and non-buildings.

Inria Aerial Image Labeling Dataset is a remote sensing image dataset of urban building scenes, including high-density metropolitan financial districts and low-density alpine resorts, with a variety of building types. The dataset consists of 360 color orthoimages (3-band RGB), covering urban residential areas in the United States and Austria. The spatial resolution is 0.3 m$^2$, with the training area of 337.5 km$^2$, a validation area of 67.5 km$^2$, a testing area is 405 km$^2$, and a total coverage area of 810 km$^2$. This article uses

remote sensing images of Austin from the Inria Aerial Image Labeling Dataset for training, where buildings are more densely located.

To adapt to the size of the memory space, the images from the WHU and Inria datasets are uniformly cropped to a size of 512 × 512 before being fed into the model for training. The ratio for the training set, validation set, and test set is 6:1:3.

To enhance model generalization, data augmentation techniques are applied to the images in the training set. The augmentation methods include random rotations, horizontal flips, vertical flips, and HSV color space transformation. There is a 50% probability that the input image will undergo rotation, horizontal flipping, and vertical flipping. In the HSV space, random perturbations are introduced to simulate variations in lighting conditions. Specifically, hue adjustment involves applying a random relative shift of ±10% to the original hue values, thereby simulating changes in light source color temperature and improving the model's adaptability to different lighting colors. Saturation is adjusted by randomly selecting a scaling factor between 0.3 and 1.7, covering extreme situations from grayscale (low saturation) to supersaturation (high saturation). Brightness adjustment is performed using a multiplication factor between 0.7 and 1.3 to simulate natural variations in light intensity, thereby improving the model's robustness to both bright and dark scenes. Data augmentation uses dynamic augmentation, which is only enabled during training and generates different augmentation samples for each round of training. As illustrated in Fig. 7, the first row displays augmented samples from the Massachusetts dataset, the second row from the WHU dataset, and the third row from the Inria dataset.

## Evaluation metrics

This article employs five mainstream evaluation metrics to quantitatively assess the model's extraction results, namely, accuracy, precision, recall, intersection over union (IoU) and F1 score. TP represents the number of building pixels predicted as true positives. TN represents the number of non-building pixels predicted as true negatives. FP represents the number of non-building pixels predicted as building pixels. FN represents the number of building pixels predicted as non-building pixels. F1 score is a widely adopted evaluation metric in machine learning for assessing classification model performance. It is especially valuable for imbalanced datasets, as it harmonizes precision and recall, thereby mitigating the limitations of relying on a single metric. The definitions of the five evaluation metrics are provided in Eqs. (11) to (15):

$$Accuracy = \frac{TP + TN}{TP + FP + TN + FN} \tag{11}$$

$$Precision = \frac{TP}{TP + FP} \tag{12}$$

$$Recall = \frac{TP}{TP + FN} \tag{13}$$

$$IoU = \frac{TP}{TP + FP + FN} \tag{14}$$

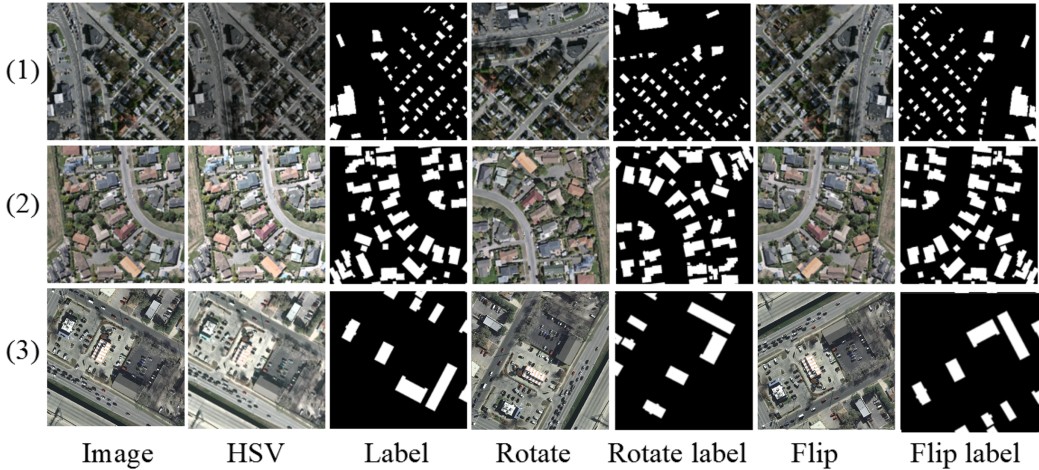

| Image | HSV | Label | Rotate | Rotate label | Flip | Flip label |

**Figure 7 Sample image of data augmentation visualization.**

$$F1 = 2 \times \frac{Precision \times Recall}{Precision + Recall}. \tag{15}$$

Building extraction is a pixel-level classification in which the cross-entropy loss function is typically used. This article selects the sigmoid cross-entropy loss as the loss function, as shown in Eqs. (16) and (17), where (i, j) represents the sample point coordinates, $y_{ij}$ indicating the true value of the sample point, $p_{ij}$ representing the probability of the true value of the sample point and logits representing the prediction results.

$$p_{ij} = Sigmoid(logits_{ij}) = \frac{1}{1 + e^{-logits_{ij}}} \tag{16}$$

$$Loss_{ij} = -[y_{ij}log(p_{ij}) + (1 - y_{ij}) \, log(1 - p_{ij})]. \tag{17}$$

## RESULT

### Results of the Massachusetts building dataset

To validate the feasibility and robustness of the method designed in this article, comparisons are made under the same configuration environment among six mainstream networks: SegNet (*Weng et al., 2020*), Deeplabv3+ (*Chen et al., 2021*), U-Net (*Peng et al., 2021*), Buildformer (*Wang et al., 2022*), SDSC-Unet (*Zhang, Zhang & Zhang, 2023*), DSATnet (*Wan et al., 2023*). To enhance the visualization effects, the extracted results are highlighted with red for false negatives (missed detections) and blue for false positives (unwarranted detections).

As shown in Fig. 8, in Scene one, due to the different spectral characteristics caused by different materials on the roofs of buildings, networks such as Segnet and Deeplabv3+ produce internal void areas in the extracted buildings, while the method proposed in this article can effectively identify buildings with different spectral characteristics. In scene two, when extracting buildings of different shapes, networks like Segnet, BuildFormer, and

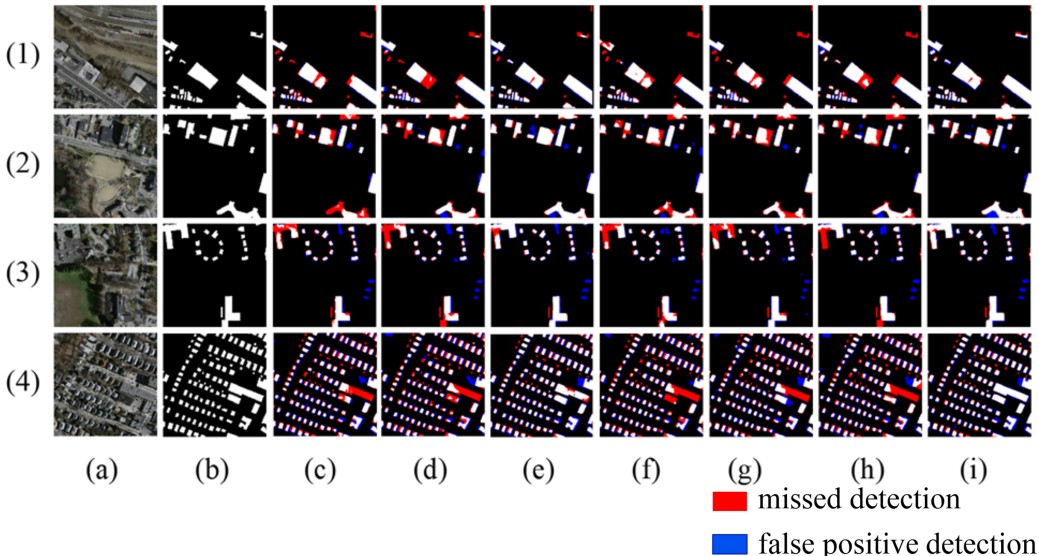

Figure 8 Extraction of Massachusetts building dataset (A) Original image; (B) Ground truth; (C) Segnet; (D) Deeplabv3+; (E) U-Net; (F) BuildFormer; (G) SDSC-Unet; (H) DSATnet; (I) Ours. The red indicates missed detections, the blue false positive detection.

SDSC-Unet show a certain degree of missed detection for buildings with complex shapes. The proposed method achieves more complete boundary extraction, which is due to the introduction of multi-scale attention gate that has a larger receptive field. In scene three, where adjacent buildings have similar grayscale to the surrounding ground, all other models have severe missed detections. In scene four, where buildings of different scales arranged densely, all other models exhibit adhesion phenomena when extracting small-scale buildings. In contrast, the proposed method extracts relatively complete boundaries for buildings in both scene three and scene four, with minimal missed detection areas.

The quantitative comparison results are shown in Table 2. From the comparison, it can be observed that the proposed method demonstrates significant improvements in all evaluation metrics compared to SegNet and DeepLabv3+. Additionally, when compared to Unet, SDSC-Unet, DSATnet, and BuildFormer, the proposed method also shows improvements in all evaluation metrics, with IoU increasing by 3.1%, 3.48%, 2% and 3.35%, F1 score increasing by 2.24%, 2.51%, 1.45%, and 2.42%, respectively. Overall, the proposed method achieves the highest performance metrics among all the compared networks.

## Results of the WHU building dataset

Figure 9 shows the extraction results for the WHU building dataset. In scene one, when the building roofs are similar to the surrounding features, Deeplabv3+ and BuildFormer fail to identify the buildings, resulting in large areas of missed detection. In scene two, the spectral and shape similarities between the buildings and roads lead to serious false detections or missed detections by other models, while the proposed method can better identify the

**Table 2 Performance evaluation results of Massachusetts building dataset.**

| Method | IoU | Precision | Recall | Acc | F1 |
|---|---|---|---|---|---|
| Segnet | 77.78 | 87.38 | 85.87 | 94.19 | 86.61 |
| Deeplabv3+ | 76.66 | 86.58 | 85.04 | 93.84 | 85.80 |
| U-Net | 76.87 | 86.07 | 85.84 | 93.80 | 85.95 |
| BuildFormer | 76.62 | 86.37 | 85.18 | 93.81 | 85.77 |
| SDSC-Unet | 76.49 | 86.68 | 84.71 | 93.83 | 85.68 |
| DSATnet | 77.97 | 87.67 | 85.83 | 94.28 | 86.74 |
| **Ours** | **79.97** | **87.72** | **88.67** | **94.71** | **88.19** |

Note:
Bold values represent the best performance indicators.

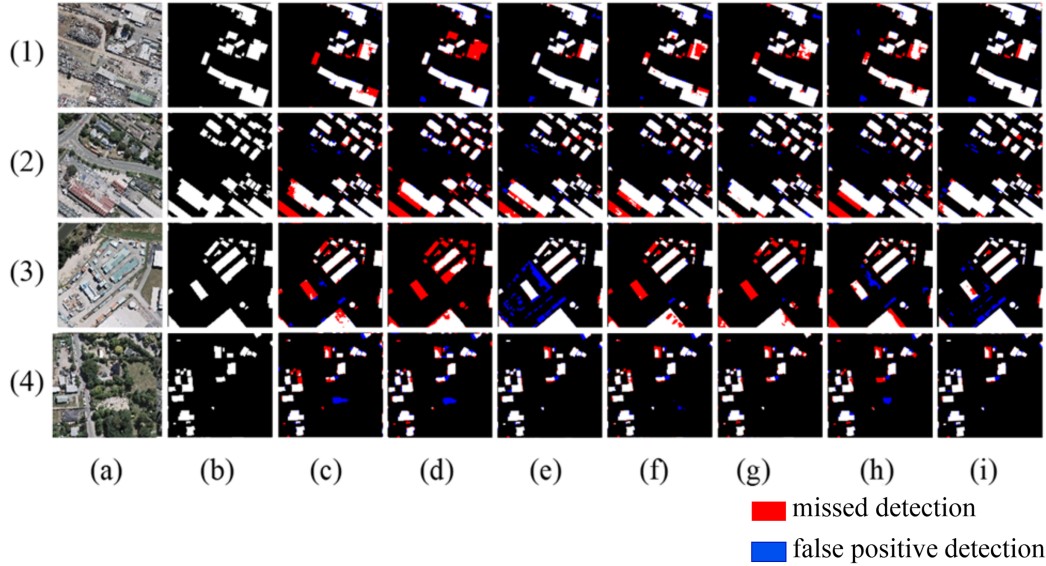

**Figure 9 Extraction of WHU building dataset (A) Original image; (B) Ground truth; (C) Segnet; (D) Deeplabv3+; (E) U-Net; (F) BuildFormer; (G) SDSC-Unet; (H) DSATnet; (I) Ours.** The red indicates missed detections, the blue false positive detections.

buildings in this scene. In scene three, small buildings are similar to surrounding containers, causing U-Net to misidentify the containers as buildings, and Segnet, Deeplabv3+, BuildFormer, and SDSC-Unet all fail to recognize buildings that resemble containers. In scene four, models such as Segnet and Deeplabv3+ misidentify small patches of open land as buildings. In all the above scenarios, the proposed method identifies the subtle differences between buildings and similar objects more effectively, enabling a more complete extraction of the buildings.

Table 3 presents the quantitative analysis results of the WHU dataset. The overall performance of the extraction results from the WHU dataset is higher than that of the Massachusetts dataset, mainly because the remote sensing images in the WHU dataset are clearer and have less interference from tree shadows. The method proposed in this article achieved scores of 93.91%, 96.42%, 97.19%, 98.53% and 96.8% for IoU, precision, recall, accuracy, and F1 score, respectively, outperforming other models across all five

**Table 3 Performance evaluation results of WHU building dataset.**

| Method | IoU | Precision | Recall | Acc | F1 |
|---|---|---|---|---|---|
| Segnet | 92.65 | 96.55 | 95.67 | 98.24 | 96.11 |
| Deeplabv3+ | 89.73 | 94.51 | 94.34 | 97.46 | 94.42 |
| U-Net | 92.73 | 95.72 | 96.59 | 98.22 | 96.15 |
| BuildFormer | 91.62 | 95.10 | 95.96 | 97.93 | 95.53 |
| SDSC-Unet | 91.88 | 95.83 | 95.54 | 98.03 | 96.08 |
| DSATnet | 91.90 | 96.09 | 95.29 | 98.04 | 95.68 |
| **Ours** | **93.91** | **96.42** | **97.19** | **98.53** | **96.80** |

Note:
Bold values represent the best performance indicators.

performance evaluation metrics. Additionally, when compared to BuildFormer, SDSC-Unet, U-Net, and DSATnet, the proposed method demonstrates superior performance, achieving higher IoU by 2.29%, 2.03%,1.18%, and 2.01%, respectively.

## Results of the Inria building dataset

Figure 10 shows the results extracted from the Inria building dataset. In scene one and scene two, some buildings are severely obscured by trees, and other models fail to fully recognize these buildings. The proposed method effectively perceives the location of buildings by enhancing positional information and strengthening the details of the location, thereby identifying the obstructed buildings more accurately. In scene three, compared with other models, the proposed method has the smallest false detection area and extracts the building boundaries the most accurately. In scene four, Segnet, U-Net, SDSC-Unet, and DSATnet show incomplete extraction of large buildings with severe internal voids. However, Deeplabv3+, BuildFormer, and the proposed method can effectively alleviate the internal voids, indicating that the multi-scale attention gate module of this article has a larger receptive field and can extract large buildings more completely.

Table 4 presents the quantitative comparison results for the Inria building dataset. Compared to the WHU building dataset, the Inria building dataset contains a richer variety of buildings with more diverse spectral features, which leads to a decrease in overall extraction results compared to the WHU building dataset. Additionally, the Inria building dataset has severe tree and shadow occlusions, demanding a higher generalization capability of the network. As shown in Table 4, the proposed method can still achieve scores of 84.83%, 89.71%, 93.23%, 96.65%, and 91.43%, for the IoU, precision, recall, accuracy and F1 score, respectively. The proposed method outperforms BuildFormer, SDSC-Unet, U-Net, and DSATnet, with IoU improvements of 1.18%, 3.04%, 2.02%, and 5.53%, and F1 score enhancements of 0.8%, 3.04%, 1.21%, and 3.71%, respectively.

We further investigate the relationship between model complexity and computational efficiency. As summarized in Table 5, our model maintains comparable parameter counts to DSATnet and Builder while requiring less memory space than both Deeplabv3+ and DSATnet. Although the proposed model exhibits relatively lower FPS (Frames Per Second) and longer processing time per sample, this trade-off is strategically designed to achieve an

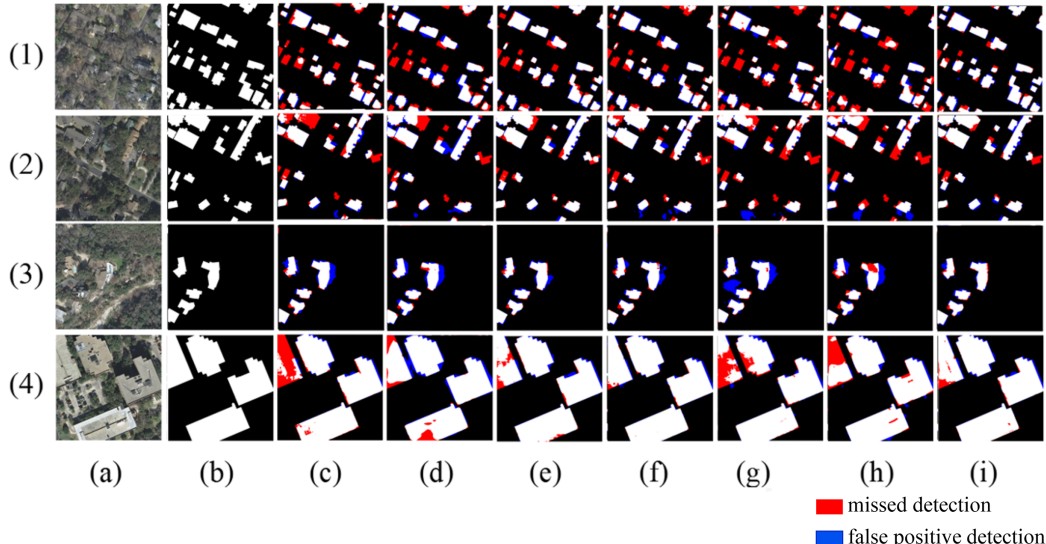

■ missed detection
■ false positive detection

**Figure 10 Extraction of Inria building dataset (A) Original image; (B) Ground truth; (C) Segnet; (D) Deeplabv3+; (E) U-Net; (F) BuildFormer; (G) SDSC-Unet; (H) DSATnet; (I) Ours.** The red indicates missed detections, the blue false positive detections.

**Table 4 Performance evaluation results of Inria building dataset.**

| Method | IoU | Precision | Recall | Acc | F1 |
|---|---|---|---|---|---|
| Segnet | 83.95 | 90.25 | 91.35 | 96.54 | 90.79 |
| Deeplabv3+ | 79.57 | 86.06 | 89.82 | 95.24 | 87.89 |
| U-Net | 82.80 | 87.56 | 93.05 | 96.04 | 90.22 |
| BuildFormer | 83.64 | 89.22 | 92.10 | 96.38 | 90.63 |
| SDSC-Unet | 80.43 | 88.16 | 88.64 | 95.67 | 88.39 |
| DSATnet | 79.29 | 85.79 | 89.73 | 95.15 | 87.72 |
| **Ours** | **84.82** | **89.71** | **93.23** | **96.65** | **91.43** |

Note:
Bold values represent the best performance indicators.

**Table 5 Comparison chart of parameter quantity and time efficiency.**

| Method | Paramete (M) | Fps | Inference tim (ms) |
|---|---|---|---|
| Segnet | 29.44 | 37.33 | 26.79 |
| Deeplabv3+ | 54.71 | 42.48 | 23.54 |
| U-Net | 34.53 | 25.49 | 39.24 |
| BuildFormer | 40.52 | 49.04 | 20.39 |
| SDSC-Unet | 21.32 | 24.33 | 41.11 |
| DSATnet | 48.37 | 33.91 | 29.49 |
| Ours | 45.36 | 10.56 | 94.72 |

optimal balance among accuracy, robustness, and generalization capability. It is worth noting that while models with higher FPS demonstrate faster inference speeds, they typically compromise either accuracy or performance, particularly in complex scenarios.

## ABLATION STUDY

To validate the effectiveness of the proposed method, ablation comparison experiments were conducted on the Massachusetts dataset under the same computer software, hardware, and parameter settings as described. The experiments compared the extraction effects of the basic model without any mechanism (U-Net), the model with Enhanced Positional Information (EPI), the model with Multi-Scale Attention Gate (MSAG), and the full method proposed in this article (U-Net + MSAG + EPI). The results of the experiments are shown in Fig. 11. The quantitative results of the ablation study are presented in Table 6.

As shown in Fig. 11, in scene one, the small-area square buildings and slender buildings within the yellow box are located around the large-scale buildings. The U-Net cannot recognize either type of building within the box. The U-Net+MSAG can recognize the small-area square buildings but cannot recognize the other slender buildings. The U-Net +EPI model, on the other hand, can recognize the slender buildings but cannot recognize the small-area square buildings. The proposed method can accurately extract both types of buildings. In scene two, the buildings within the yellow box are interfered with by trees, shadows, vehicles, and other disturbances. U-Net shows significant adhesion issues in the extracted buildings, while U-Net+EPI fails to suppress the afore-mentioned noise, leading to an increase in the false detection rate. U-Net+MSAG, with its gating units for adjustment, can effectively reduce interference and significantly decrease the area of false detections, but the details of the extracted building edges remain incomplete. The proposed method can effectively suppress noise interference and completely extract the edge details of the buildings, resulting in more complete building extractions. In scene three, the two buildings within the yellow box are elongated and narrow, with spectral features similar to the background of the image. As a result, they cannot be fully recognized by the U-Net, U-Net+MASG, and U-Net+EPI models. The method of this article fully leverages the advantages of both mechanisms and can accurately identify the two buildings that the other three methods fail to recognize.

From the quantitative comparison results in Table 6, it can be seen that U-Net+GASM shows a more significant improvement in recall compared to U-Net, which is attributed to its ability to capture buildings at multiple scales, thereby increasing recall. U-Net+EPI exhibits a more notable enhancement in IoU, which is due to the model's capacity to learn more positional information and spatial features, resulting in a higher degree of overlap with the ground truth. The proposed method combines the advantages of the two afore-mentioned approaches, demonstrating better overall performance. Compared to U-Net, the five performance metrics are improved by 3.1%, 1.65%, 2.83%, 0.91% and 2.24%, respectively.

To verify the impact of each module on all layers of the network, the ablation study was improved by increasing the parameters of all layers. As shown in Table 7, the MSAG and EPI modules are gradually added between different layers. In groups 2-5, the performance

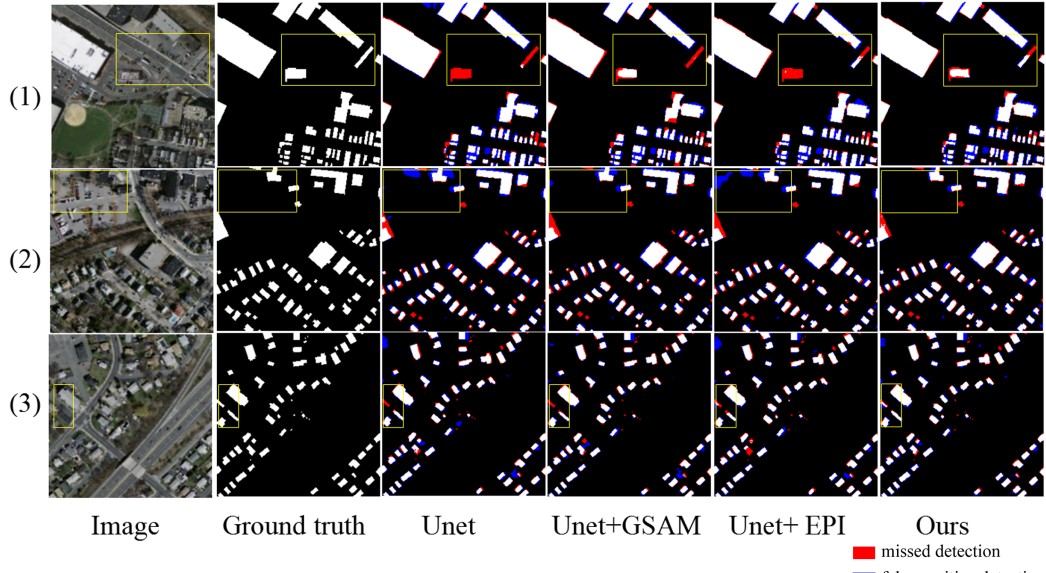

**Figure 11 Results of ablation experiment extraction.** The red areas indicate missed detections, while the blue areas represent false positive detections.

**Table 6 Performance evaluation results of ablation experiments.**

| Method | IoU | Precision | Recall | Acc | F1 |
|---|---|---|---|---|---|
| U-Net | 76.87 | 86.07 | 85.84 | 93.80 | 85.95 |
| U-Net+MSAG | 79.34 | 87.10 | 88.43 | 94.48 | 87.75 |
| U-Net+ EPI | 79.61 | 87.41 | 88.30 | 94.58 | 87.85 |
| **Ours** | **79.97** | **87.72** | **88.67** | **94.71** | **88.19** |

Note:
Bold values represent the best performance indicators.

**Table 7 Table of ablation studies with different layer parameters.**

| Group | Different layers | | | | Module | | Evaluation | | | | |
|---|---|---|---|---|---|---|---|---|---|---|---|
| | Layer1 | Layer2 | Layer3 | Layer4 | MSAG | EPI | Iou | Precision | Recall | Acc | F1 |
| 1 | | | | | | | 76.87 | 86.07 | 85.84 | 93.80 | 85.95 |
| 2 | √ | | | | √ | | 76.96 | 87.03 | 85.05 | 93.97 | 86.02 |
| 3 | √ | √ | | | √ | | 77.56 | 87.57 | 85.40 | 94.17 | 86.47 |
| 4 | √ | √ | √ | | √ | | 78.06 | 87.62 | 86.05 | 94.29 | 86.82 |
| 5 | √ | √ | √ | √ | √ | | 79.34 | 87.10 | 88.43 | 94.48 | 87.75 |
| 6 | | √ | | | | √ | 75.05 | 87.07 | 82.44 | 93.58 | 84.69 |
| 7 | | √ | √ | | | √ | 79.35 | 87.67 | 87.81 | 94.56 | 87.73 |
| 8 | | √ | √ | √ | | √ | 79.61 | 87.41 | 88.30 | 94.58 | 87.85 |

of the model gradually improves as the MSAG module is incrementally added to the base architecture. Compared with Group1, Group 5 demonstrates significant improvements, with the IoU, recall, and F1 scores increasing by 2.47%, 2.59%, and 1.8%, respectively. Group 6 only adds the EPI module in a single level, with only precision higher than U-Net at 87.07%, indicating that the EPI module improves the discriminative power of positive samples, but its IoU decreased by 1.82% and recall decreased by 2.61%, indicating that the model retains more details while being affected by some noise interference. In Group 7, the addition of EPI modules to Layer 2 and Layer 3 leads to a notable enhancement in performance compared to Group 6, with the IoU, Recall, and F1 Score reaching 79.35%, 87.81%, and 87.73%, respectively. Further extending this configuration, Group 8 incorporates EPI modules into Layer 2, Layer 3, and Layer 4, achieving even higher performance metrics, with IoU, recall, and F1 scores of 79.61%, 88.47%, and 87.93%, respectively. The results indicate that the EPI modules collaborate with each other at every layer of the network, and adding only EPI modules in a single layer cannot fully utilize the functions of the modules. It is necessary to add EPI modules in more layers to better enhance the feature extraction ability of the model.

To validate the effectiveness of the multi-scale attention gate module in enhancing the model's receptive field, we performed ablation studies using images from three datasets. The resulting heatmaps are presented in Fig. 12, where panels (a), (b), and (c) display samples from the Massachusetts, WHU, and Inria datasets, respectively. The "Encoding stage" represents the heatmap output by the U-Net during the downsampling encoder stage, "MSAG" refers to the attention-enhanced encoding layer outputs. It can be observed that integrating the attention mechanism increases global information weights in heat maps. Unlike standard downsampling U-net convolutions limited to local information extraction, our multi-scale attention gate has a larger receptive field, which can effectively establish global dependencies and have better perception for multi-scale structures.

## DISCUSSION

By comparing with mainstream network models, the proposed method demonstrates higher accuracy and stability, reflecting the superiority of the approach. The introduced multi-scale attention gate module effectively expands the receptive field, thereby enabling a more comprehensive extraction of multi-scale features from the image. The added positional information enhancement module reduces the loss of multi-scale information in the traditional upsampling method during the decoding process. Meanwhile, during the fusion of high and low-level features, it effectively captures the positional information of buildings, further strengthening the positional information of the features. Combining the relative advantages of these two modules, the proposed method can significantly improve the extraction capability of multi-scale buildings. Especially when processing remote sensing images with complex backgrounds, the improvements of the proposed method optimize the details such as building boundaries and corners, allowing for more accurate localization and extraction of building shapes and edges.

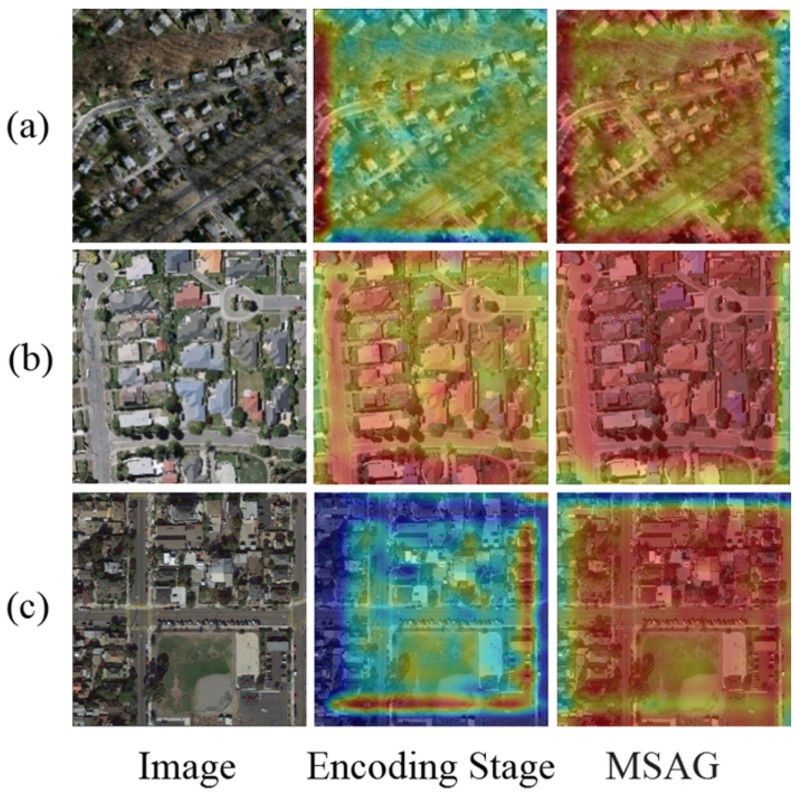

**Figure 12  Heat map comparison.**         

## CONCLUSION

To address issues such as edge blurring, incomplete structure, and loss of detail in building extraction with traditional U-shaped networks, this article proposes a novel method integrating multi-scale attention gates and enhanced positional information. The method is validated on three public datasets (Massachusetts, WHU, and Inria) and compared with six state-of-the-art approaches. Experimental results demonstrate that the proposed method effectively integrates multi-scale features, achieving superior accuracy and preserving more complete building edges and details. Ablation experiments are conducted to analyze module performance, revealing that the proposed modules significantly improve localization accuracy and edge information extraction by leveraging comprehensive multi-scale feature integration. The proposed method demonstrates excellent performance across multiple datasets, exhibiting strong generalization capabilities and practical applicability under diverse geographical conditions. It provides reliable technical support for digital urban planning, urban expansion analysis, and map production by effectively handling building extraction in complex scenarios. However, the proposed method involves a large number of parameters, which leaves room for further optimization in terms of computational efficiency. Furthermore, while validated on public datasets, its generalizability to real-world engineering scenarios requires further verification. Future work will explore multi-source data fusion strategies (*e.g.*, light detection and ranging (LiDAR) point clouds, infrared bands, and OpenStreetMap vector

data) to establish cross-modal feature complementation mechanisms, thereby enhancing segmentation robustness in building-background ambiguous regions. Additionally, future work will carry out lightweight design, such as using depthwise separable convolutions, to achieve deployment on mobile devices and real-time systems, meeting practical application needs.

### Funding
This work was supported by the Scientific Research Startup Foundation of Fujian University of Technology (No. GY-Z24009) and the Open Project of Fujian Key Laboratory of Spatial Information Perception and Intelligent Processing (Yango University, No. FKLSIPIP1020). The funders had no role in study design, data collection and analysis, decision to publish, or preparation of the manuscript.

### Grant Disclosures
The following grant information was disclosed by the authors:
Scientific Research Startup Foundation of Fujian University of Technology: GY-Z24009.
Open Project of Fujian Key Laboratory of Spatial Information Perception and Intelligent Processing (Yango University): FKLSIPIP1020.

### Competing Interests
The authors declare that they have no competing interests.

### Author Contributions
- Rui Xu conceived and designed the experiments, performed the experiments, analyzed the data, performed the computation work, prepared figures and/or tables, authored or reviewed drafts of the article, and approved the final draft.
- Renzhong Mao conceived and designed the experiments, performed the experiments, analyzed the data, performed the computation work, prepared figures and/or tables, authored or reviewed drafts of the article, and approved the final draft.
- Zhenxing Zhuang conceived and designed the experiments, performed the experiments, analyzed the data, performed the computation work, prepared figures and/or tables, and approved the final draft.
- Fenghua Huang analyzed the data, performed the computation work, prepared figures and/or tables, authored or reviewed drafts of the article, and approved the final draft.
- Yihui Yang analyzed the data, prepared figures and/or tables, and approved the final draft.

### Data Availability
The raw data is available at Building Image Dataset and Zenodo:
- https://pan.fjut.edu.cn/s/JkNXYFs686QpsaT

- mao, . renzhong. (2025). Inria, WHU, Massachesetts Building Dataset [Data set]. Zenodo. https://doi.org/10.5281/zenodo.15089910.

Third-party datasets:
- https://www.cs.toronto.edu/%7Evmnih/data/
- http://gpcv.whu.edu.cn/data/building_dataset.html
- https://project.inria.fr/aerialimagelabeling/

The code is available in the Supplemental Files.

## Supplemental Information

Supplemental information for this article can be found online at http://dx.doi.org/10.7717/peerj-cs.2826#supplemental-information.

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
