# Peer review of "Building extraction from remote sensing images based on multi-scale attention gate and enhanced positional information"

_PeerJ Computer Science, doi:10.7717/peerj-cs.2826_

## Round 0.1 · original submission · Major Revisions

Dear authors,

Your paper has been reviewed. Based on the reviewers' report, major revisions are required before it can be considered for publication in PEERJ Computer Science. Specifically, the following issues need to be addressed:

1) Considering that whether the ablation experiment can be carried out depends on the architecture of the actual model and that not all components can be carried out, the author has initially achieved the goal. But in terms of the final result, the modifications that the author claims have advantages are not shown in the results because the overall results are very similar to Unet. The authors must clarify this point.
2) The authors used a relatively small test dataset, which may have led to artificially high accuracy that lacks credibility. The authors must clarify this point.
3) Some information in the abstract could be more specific. For instance, explicitly name the datasets and comparative models used. It is also recommended that the overall performance improvements be presented in terms of specific evaluation metrics and numerical values.
4)The conclusions are generic. It is recommended that the authors systematically elaborate on the proposed method's limitations, potential impact on practical applications (e.g., urban planning), and future research directions.

·

Basic reporting

The English expression is basically qualified, and the author's experimental ideas and procedures can be understood.

Experimental design

Considering that whether the ablation experiment can be carried out depends on the architecture of the actual model, and not all components can be carried out, the author has initially achieved the goal. But in terms of the final result, the modifications that the author claims have advantages are not shown in the results, because the overall results are very similar to Unet.

Validity of the findings

Since PeerJ does not evaluate the influence and innovation of articles in this field, I can only comment on the data process. Since the performance of the modified custom network is not much higher than the original Unet architecture, not even 2%, I have a hard time agreeing that this result is "better than the current literature".

Additional comments

none

·

Basic reporting

The manuscript is organized effectively, featuring a comprehensive introduction, literature review, methodology, experiments, results, and discussion. It tackles shortcomings in conventional extraction methods by incorporating innovative modules such as MSAG and improved positional information in U-Net. The experiments were validated using three datasets (Massachusetts, WHU, and Inria), showcasing better performance compared to leading methods like SegNet, Deeplabv3+, and BuildFormer.
Metrics include IoU, Precision, Recall, and Accuracy, with notable improvements in all benchmarks compared to baseline methods.

Experimental design

The methods are described in enough detail to enable replication:
The network structure is clearly outlined, accompanied by diagrams that illustrate the architecture and module designs (e.g., Figures 1–5).
Mathematical formulations for the MSAG and EPI modules are included, explaining feature fusion, attention mechanisms, and how positional information is encoded.
Hyperparameters (such as learning rate, batch size, and optimizer settings) along with the training environment (Ubuntu system, Pytorch framework, GPU specifications) are clearly specified.
Preprocessing steps, including cropping and dataset splits (training:validation:test = 8:1:1), are thoroughly documented.
While the methods are thorough, offering more details on the hyperparameter tuning process or the criteria for training convergence would improve replicability. Including code or implementation details, such as a GitHub repository, would also aid in supporting reproducibility. Additionally, a short discussion on computational efficiency, including runtime and memory usage, would be useful considering the resource-intensive nature of deep learning models.

Validity of the findings

While the authors address the generalizability of the model, offering more insights into its limitations, such as computational costs and scalability to larger datasets, would lead to a more balanced conclusion. Including raw data or result outputs, like segmentation masks, as supplementary materials would also improve transparency. A short discussion on potential future directions, such as integrating other architectural features like transformers or hybrid models, would be beneficial.

Additional comments

The suggested improvements encompass:
1. Offering more in-depth information on hyperparameter tuning and the criteria for training convergence.
2. Incorporating raw data or additional segmentation results to enhance transparency.
3. Improving the ablation study by adding parameters of all the layers.
4. Discussing computational efficiency and potential limitations to ensure a well-rounded conclusion.
5. Emphasizing future research directions that could expand upon the current study.

·

Basic reporting

The paper is generally well-structured, and the experiments are comprehensive. However, there is room for improvement in writing and certain details. The following suggestions are proposed:
1. In abstract, some information could be more specific. For instance, explicitly name the datasets and comparative models used. It is also recommended to present the overall performance improvements in terms of specific evaluation metrics and numerical values.
2. In lines 66–89, numerous network models are mentioned. For each application of building boundary segmentation, it is suggested to briefly explain their advantages and limitations to highlight the improvements brought by the proposed method.
3. Although the Introduction addresses the limitations of traditional U-Net, it is recommended to include more background information on comparative methods to emphasize the applicability and innovation of the proposed approach.
4. For Figures 6, 7, 8, and 9, it would be helpful to specify what the different colors represent in the figure captions or add a legend. Figures in academic papers should convey accurate information even without referring to the main text.
5. In Dataset section: As this is an image-processing-based paper, it is suggested to visualize the datasets to help readers quickly understand the content. The random cropping process should also be visualized or described in detail.

Experimental design

For Network structure, while the descriptions of the multi-scale attention gate module and enhanced positional information module are detailed, the definitions of the mathematical symbols in lines 170–193 are not sufficiently clear. Considering the specific definitions of convolutional kernel sizes, it is recommended to add a symbol explanation table to assist readers. Additionally, the rationale for the network parameters should be clarified.

Validity of the findings

1. For Tables 1–4, it is suggested to include statistical tests to enhance the credibility of the results.

2. The conclusions are somewhat generic. It is recommended to systematically elaborate on the limitations of the proposed method, its potential impact on practical applications (e.g., urban planning), and future research directions.

Reviewer 4 ·

Basic reporting

Figures 6-10 are missing legends.

Experimental design

The manuscript presented a building extraction method based on UNet, integrating multi-scale attention gate modules and positional information enhancement modules. By conducting experiments, the proposed network has been proved to have better performance on three public data sets, i.e., Massachusetts Building Dataset, WHU Building Dataset, and Inria Aerial Image Labeling Dataset. The authors have done some work, but the outcome is inadequate in terms of methodology and experiment. My main concerns include:
-The manuscript lacks substantial innovation, the contribution of the manuscript is mainly a combination of existing deep learning modules. The modules used, including multi-scale attention gate modules and UNet, were proposed earlier and the backbone used was not the state-of-the-art network.
-The experimental design is flawed, and the experiment results is not convincing. There are three concerns:
i) An 8:1:1 ratio of training sets, validation sets, and test sets is not reasonable. Typically, the test set should have a larger proportion. It is recommended to refer to the common split ratio of the data set used
ii) The manuscript uses a relatively small test dataset, which may lead to an artificially high accuracy that lacks credibility. For example, the IoU scores of methods DSAT-Net, and BuildFormer on the same Massachusetts building dataset in this manuscript is higher than the scores reported in Paper [1]., with IoU scores of 80.31 and 78.49, whereas Paper [2] reports only 76.54, and 75.74.
iii) Table 4 shows that the proposed method achieves some improvement through ablation experiments, but the improvement is not significant. The increase in IoU compared to the original UNet is less than 0.6%, which is negligible and could potentially be attributed to chance.
[1]Zhang, R., Wan, Z., Zhang, Q., & Zhang, G. (2023). DSAT-net: Dual spatial attention transformer for building extraction from aerial images. IEEE Geoscience and Remote Sensing Letters.

Validity of the findings

no comment

Additional comments

no comment

---

## Round 0.2 · Minor Revisions

Dear Authors,
Your paper has been revised. It needs minor revisions before being accepted for publication in PEERJ Computer Science. More precisely

1) The manuscript lacks a clear explanation of the proportion of augmented data in the dataset partition, which is a crucial aspect. Further discussion on data augmentation methods may be necessary.

2) The manuscript does not specify whether the flipping transformation includes both horizontal and vertical flipping, which should be clarified in the data augmentation section.

3) The manuscript does not clearly indicate whether the augmented data is added to the total dataset or used as a replacement. Furthermore, it does not specify whether the augmented data is only used for the training set or included in all sets, nor does it clarify the proportion of augmented data in the dataset.

4) The revised manuscript mentions the data augmentation strategy, but the current approach appears somewhat simplistic. Further discussion on data augmentation methods must be added.

·

Basic reporting

The revised abstract more intuitively conveys the achieved performance improvements. The Introduction section now better highlights the research motivation and innovations through a more detailed discussion of U-Net-related methods. Table 1 has enhanced the readability of the paper. The figures have been improved by adding legends. However, in Figure 7, the Flip illustration in the third row contains an error and needs correction. The availability of data links and open-source code ensures the reproducibility of the research.

Experimental design

The revised manuscript mentions the data augmentation strategy, but the current approach appears somewhat simplistic. Further discussion on data augmentation methods may be necessary. Additionally, the manuscript lacks a clear explanation of the proportion of augmented data in the dataset partition, which is a crucial aspect.

Validity of the findings

no comment

Additional comments

1. Figure 7, the Flip illustration in the third row is incorrect. Moreover, the manuscript does not specify whether the flipping transformation includes both horizontal and vertical flipping, which should be clarified in the data augmentation section.
2. The current data augmentation approach is too simplistic and requires further discussion.
3. The usage of the augmented dataset needs to be explicitly stated. The manuscript does not clearly indicate whether the augmented data is added to the total dataset or used as a replacement. Furthermore, it does not specify whether the augmented data is only used for the training set or included in all sets, nor does it clarify the proportion of augmented data in the dataset.

---

## Round 0.3 · Minor Revisions

Dear Authors,
Your paper has been revised. Minor revisions are needed before it is accepted for publication in PEERJ Computer Science. More precisely:

1) The authors did not identify the implementation of the EPI (Enhanced Position Information) mechanism in the code described in the manuscript.

2) According to the data and comparison results in the paper, the proposed method has limited improvement compared to U-Net and other SOTA methods, but it does perform better in certain specific scenarios (such as multi-scale buildings and occluded buildings). The authors need to state this point more clearly in their manuscript.

·

Basic reporting

All relevant information is complete and comprehensive.

Experimental design

1. I have reviewed the provided code to ensure that MSAG (Multi-Scale Attention Gating) is implemented and applied in the model, aligning with the main technical contributions of the paper.

2. The implementation of the EPI (Enhanced Position Information) mechanism was not clearly identified in the code.

3. The paper conducted an ablation study and tested the impact of different components on model performance.

Validity of the findings

1. Compared to the basic U-Net, IoU, Precision, Recall, Accuracy, and F1-score increased by 3.1%, 1.65%, 2.83%, 0.91%, and 2.24% respectively.

2. The main improvements of the paper lie in Multi-Scale Attention Gating (MSAG) and Enhanced Positional Information (EPI), but some concepts mentioned, such as skip connections and multi-layer feature fusion, are actually already present in U-Net itself. If there are minor modifications later, consider reducing the description of features that U-Net already has and clarify them explicitly.

3. According to the data and comparison results in the paper, this method has limited improvement compared to U-Net and other SOTA methods, but it does perform better in certain specific scenarios (such as multi-scale buildings and occluded buildings).

Additional comments

According to the discussion and conclusion of the paper, the author indeed mentioned the limitations of the method, but overall emphasized its contributions while mentioning less about the limited scope for improvement.

·

Basic reporting

no comment

Experimental design

no comment

Validity of the findings

no comment

Additional comments

After revision, the manuscript has been more clearly clarified about the data augmentation scheme in the training phase. The erroneous images have been corrected. It is now recommended for acceptance.

---

## Round 0.4 · accepted · Accept

Dear Authors,
Your paper has been revised. It has been accepted for publication in PEERJ Computer Science. Thank you for your fine contribution.

·

Basic reporting

Reasonable explanations have been made and corresponding modifications provided. It can be accepted in its current form.

Experimental design

No modification required

Validity of the findings

No modification required